# Magnetic Properties of FeNi_3_ Nanoparticle Modified *Pinus radiata* Wood Nanocomposites

**DOI:** 10.3390/polym11030421

**Published:** 2019-03-05

**Authors:** LiLi Wang, Na Li, Tiqi Zhao, Bin Li, Yali Ji

**Affiliations:** College of Science, Northeast Forestry University, Harbin 150040, China; aichikfc.chan@163.com (N.L.); zhao_scaler@163.com (T.Z.); libinrainkiss1014@163.com (B.L.); jyl1513062809@163.com (Y.J.)

**Keywords:** FeNi_3_ nanoparticles, natural polymer, modified wood nanocomposites, magnetism, saturation magnetization

## Abstract

Magnetic FeNi_3_ nanoparticles were synthesized in the internal structure of wood through an in situ fabrication approach. The morphology, crystalline phase and chemical composition of the FeNi_3_ modified wood was investigated by X-ray powder diffraction (XRD), Fourier transform infrared (FTIR) spectroscopy, scanning electron microscopy (SEM) with energy-dispersive X-ray (EDX) analysis and X-ray photoelectron spectroscopy (XPS). SEM confirmed that the magnetic nanoparticles were densely dispersed in the wood matrix. The magnetic hysteresis loops showed that the magnetism of composites is dependent on the amount of FeNi_3_ loading. The saturation magnetization of magnetic wood increases from 6.3 to 10.8 emu/g with an increase of FeNi_3_ loading from 12 to 18 wt %. Furthermore, magnetic wood showed significant directional dependence. The presented work will provide a feasible pathway for producing wood composite products.

## 1. Introduction

Wood is a structured and ordered natural polymer material, which has excellent hierarchical structures and physical properties due to its natural growth [1]. It is used in several important engineering applications, such as interior decoration, building materials and in furniture industry. Furthermore, wood with a microporous structure can combine a natural porosity scaffold with inorganic/organic compounds. Therefore, many efforts have been devoted to creating functionalized wood by dispersing inorganic and organic components into the wood matrix in order to endow these wood-based materials with new properties, such as biodegradation resistance [2], rheological properties [3], mechanical properties [4,5,6], acoustic properties [7], and magnetic properties [8]. Among these properties, magnetism has recently received much attention.

Magnetic wood is a composite that exhibits the inherent lightweight properties of the wood substrate and the magnetic properties of nanoparticles, which achieves a good harmony between both wood and magnetism [9]. Thus, magnetic wood has potential applications in electromagnetic shielding, as an indoor electromagnetic wave absorber, as a heating board and for heavy metal adsorption [10,11,12,13]. So far, studies on magnetic wood have mostly focused on ferrite particle modified wood, such as Fe_3_O_4_ [14], CoFe_2_O_4_ [15], MnFe_2_O_4_ [16] and so on. Fe, Ni and their alloys are very important soft magnetic materials due to their higher saturation magnetization and smaller coercive forces when compared with ferrites [17,18,19,20]. Furthermore, they have a wide range of applications, such as acting as electromagnetic wave absorbers, transformers, magnetic sensors and magnetic recording heads [21,22]. 

When the molar ratio of Fe to Ni is 1/3, the FeNi_3_ intermetallic compound can be obtained. Lu et al. reported that ferromagnetic FeNi_3_ nanoparticles have been successfully synthesized via a hydrazine hydrate reduction in aqueous solution at room temperature. Furthermore, the saturation magnetization reaches about 110 emu/g [23]. In addition, Yuan et al. reported that the FeNi_3_ alloy was prepared using hydrazine hydrate as a reducing agent in strong alkaline media. The saturation magnetization of FeNi_3_ alloy is 48.48 emu/g [24]. However, FeNi_3_ nanoparticle modified wood composites have not been reported until now. Hydrazine hydrate is a low-cost moderate reducing agent and has been proven to be good alternative to produce FeNi_3_ metallic particles. Therefore, in this study, FeNi_3_ nanoparticles were formed in the cell cavity of wood using hydrazine hydrate as a reducing agent through a simple in situ synthesis. The morphology, chemical composition and magnetism of the obtained modified wood composites was investigated.

## 2. Materials and Methods

### 2.1. Materials

*P. radiata* pine specimens were cut into blocks of 20 mm (longitudinal) × 20 mm (radial) × 10 mm (tangential) and dried in a vacuum oven (DHG-9203A, Shanghai, China) at 60 °C for 12 h to a constant weight. All the reagents in the experiment were analytical grade and used without further purification. Ferrous ammonium sulfate hexahydrate ((NH_4_)_2_Fe(SO_4_)_2_·6H_2_O), nickel sulfate hexahydrate (NiSO_4_·6H_2_O), sodium hydroxide (NaOH), hydrate hydrazine (N_2_H_4_·H_2_O, 80%) and absolute ethanol (C_2_H_5_OH) were purchased from the Tianjin Kemiou Fine Chemical Reagent Co. (Tianjin, China). Deionized water (Milli-Q Academic, Beijing, China) was exclusively used in all aqueous solutions and rinsing.

### 2.2. Preparation of FeNi_3_ Nanoparticle Magnetic Wood Composites

The mixture of (NH_4_)_2_Fe(SO_4_)_2_·6H_2_O and NiSO_4_·6H_2_O with a molar ratio of 1/3 was dissolved in 50 mL distilled water to form three homogeneous solutions: A, B and C, respectively. Mixed solution A, with a concentration of 0.6 mol/L, contains 2.94 g (NH_4_)_2_Fe(SO_4_)_2_·6H_2_O (0.0075 mol) and 5.91 g NiSO_4_·6H_2_O (0.0225 mol). Mixed solution B, with a concentration of 0.8 mol/L, contains 3.92 g (NH_4_)_2_Fe(SO_4_)_2_·6H_2_O (0.01 mol) and 7.88 g NiSO_4_·6H_2_O (0.03 mol). Mixed solution C, with a concentration of 1.0 mol/L contains 4.90 g (NH_4_)_2_Fe(SO_4_)_2_·6H_2_O (0.0125 mol) and 9.86 g NiSO_4_·6H_2_O (0.0375 mol). Firstly, the wood specimens were entirely dipped into solutions A, B, and C under a lowered pressure of 0.09 MPa for 2 h, respectively. NaOH (2 g) and hydrazine hydrate (9.0 mL) were added in 50 mL distilled water to form an alkaline hydrate hydrazine solution. Subsequently, the above pretreated wood samples were rapidly soaked into the alkaline hydrate hydrazine solution with a mole concentration of 3 mol/L at 70 °C for 6 h. The formed nanoparticles were labeled as FeNi_3_. Finally, FeNi_3_ modified magnetic wood composites were washed with distilled water and absolute ethanol three times and dried in a vacuum drying oven at 60 °C.

The weight percent gains (WPGs) (%) of the wood samples after treatment were calculated by the following equation:WPG (%) = (*W*_1_ − *W*_0_)/*W*_0_ × 100 %(1)
Where *W*_0_ and *W*_1_ are the weights of the oven dried wood before and after treatment, respectively. 

The WPGs for the three modified wood samples were 12, 15 and 18 wt % at 0.6, 0.8 and 1.0 mol/L ion concentrations of the precursor solution, respectively. Thus, the resulting specimens of the modified wood were labelled as MW–12 wt %, MW–15 wt % and MW–18 wt %, respectively. Furthermore, the experimental scheme for the preparation of the FeNi_3_ nanoparticle modified wood is displayed in Figure 1.

### 2.3. Characterization Techniques

The morphology of the unmodified wood and modified wood composites was observed through scanning electronic microscopy (SEM, Quanta 200, FEI Company, Eindhoven, Holland). Chemical compositions of the FeNi_3_ modified wood composites were detected by energy-dispersive X-ray spectroscopy (EDX, EDAX Inc., Mahwah, NJ, USA). The crystalline structure of unmodified wood and modified wood were identified using the X-ray diffraction technique (XRD, D/MAX–3B, Rigaku, Tokyo, Japan), using Cu Kα radiation (λ= 1.5418 Å), a generator operated at 1,200 W (40 kV× 30 mA), and a 2θ scan range from 10° to 80°, with a scanning speed of 4°/min. For the Fourier transformation infrared (FTIR, Thermo Nicolet, Madison, WI, USA) spectroscopy, thin sample disks were separately made by grinding small portions of the wood specimens and pressing them with potassium bromide, then the spectra of the unmodified and modified wood were obtained, with a scanning range of 4000–500 cm^−1^ at a resolution of 4 cm^−1^. X-ray photoelectron spectroscopy (XPS, K–Alpha, Thermo Fisher Scientific Company, Waltham, MA USA) was used for the surface analysis of the chemical composition of modified wood. The hysteresis loops of the samples were investigated using a vibrating sample magnetometer (VSM, MPMS–XL–7, Quantum Design Company, San Diego, USA) with a maximum applied magnetic field of 10,000 Oe at room temperature (about 300 K). 

## 3. Results and Discussion

### 3.1. XRD Analysis of Unmodified and Modified Wood

Figure 2a–d shows the XRD patterns of the unmodified wood and the three modified wood samples, respectively. From Figure 2a, the diffraction peaks of the unmodified wood at 16° and 22° correspond to crystal planes (101) and (002) of cellulose, respectively [25]. For all of the modified woods, these characteristic peaks of pristine wood are distinctly weakened, shown in Figure 2b–d, which is due to the deposited metal particles destroying the crystalline structure of the cellulose, resulting in a decrease in the crystalline structure of the cellulose. Moreover, characteristic peaks at 2*θ* values of 44.3°, 51.7° and 75.8° correspond to the crystal planes of (111), (200), and (220) of the FeNi_3_ particles, respectively [26]. This is the typical diffraction pattern of a face-centered cubic (FCC) phase (JCPDS No. 38–0419). Meanwhile, the WPGs of the magnetic wood composites increase from 12 to 18 wt % with an increase of ion concentration from 0.6 to 1.0 M. This indicates that the ferrous and nickel ion concentrations can control the content of the FeNi_3_ nanoparticles in the wood composites. Thus, the intensity of the diffraction peak increases obviously with the increase of FeNi_3_ loading. The above analysis indicates that FeNi_3_ particles with high crystallinity and homogeneity are successfully prepared in wood.

### 3.2. FTIR Analysis of Unmodified and Modified Wood

Figure 3 shows the FTIR spectra of unmodified wood and MW–18 wt %. For the unmodified wood, a strong O–H stretching absorption at 3430 cm^−1^ is observed. The bands at 2930, 1730, 1650 and 1510 cm^−1^ are attributed to the stretching vibration of C–H in CH_3_, stretching vibrations of C=O in hemicelluloses, stretching vibrations of conjugated carbonyl and aromatic stretching vibrations of lignin, respectively [27,28,29]. For the modified wood, the characteristic absorption bands of wood have changed when compared with the unmodified wood. The bands at around 1730 cm^−1^ have disappeared because of the degradation of acetyl groups in hemicelluloses by magnetic treatment [30]. Meanwhile, the O–H stretching absorption band at 3430 cm^−1^ have shifted to a lower wavenumber, 3400 cm^−1^, confirming that the interactions occurred between the OH groups of the wood matrix and the formed FeNi_3_ nanoparticles through hydrogen bonds [31]. A new characteristic band at 611 cm^-1^ is attributed to O–Fe and O–Ni stretching modes [32,33]. Moreover, the bands intensities at around 1590 cm^−1^ and 1440 cm^−1^ were significantly enhanced, which can be attributed to chemical bonding between the surface-active hydroxyl groups of the FeNi_3_ nanoparticles and the carboxyl groups of the wood [34].

### 3.3. Morphology Analysis of Unmodified and Modified Wood Composites

Figure 4a–d shows the SEM images of longitudinal cross-sections of the unmodified wood and MW–18 wt %. From Figure 4a, it can be seen that the unmodified wood has microgrooved structures and smooth lumen walls. However, the modified wood composites show that the microgrooved structures of the original wood have been covered by modified particles, as seen in Figure 4b. This reveals that the FeNi_3_ particles are adhering to the lumen walls of the wood. As shown in Figure 4c, a large amount of FeNi_3_ particles are deposited together continuously and appear as a densely deposited layer. Figure 4d indicates that the morphology of the FeNi_3_ nanoparticles is a rounded disk, with diameter of about 100 nm. The above SEM tests confirm that the FeNi_3_ nanoparticles are imbedded into the inner cavity of wood by in situ chemosynthesis. Figure 5 shows the energy-dispersive X-ray (EDX) analysis of the modified wood. It illustrates that the atomic ratio of iron to nickel is 1:2.70, which is close to the initially set ratio of 1:3.

### 3.4. XPS Analysis of Unmodified Wood and MW–18 wt % Composite

Figure 6a,b show wide scan XPS spectra for the unmodified wood and MW–18 wt %, respectively. The major elements for the unmodified wood are C and O, as seen in Figure 6a. The peaks at 285.3 and 530.6 eV are attributed to C 1s and O 1s, respectively. Moreover, C, O, Ni and Fe elements were detected in the modified wood, as seen in Figure 6b. Figure 6c shows the XPS spectrum of the Fe 2p of the modified wood. The binding energy for the 2p3/2 peak of iron (Fe^0^) is around 706.5 eV. Furthermore, peaks centered at 709.20, 722.41, 711.14 and 724.45 eV are 2p3/2, 2p1/2 of Fe^2+^ and 2p3/2, 2p1/2 of Fe^3+^, respectively [35].

Figure 6d shows the XPS spectrum of Ni 2p. The binding energy at 852.4 and 870.0 eV is 2p3/2 and 2p1/2 of Ni^0^ [36]. Peaks at 855.2 and 872.8 eV are assigned to 2p3/2 and 2p1/2 of Ni^2+^, respectively. The binding energy at 862.0 and 878.7 eV corresponds to satellite features of 2p3/2 and 2p1/2 of Ni^2+^ [37]. The above XPS results demonstrate that Ni^2+^ and Fe^2+^ have been well reduced to Fe^0^ and Ni^0^. Furthermore, a part of the oxidation states of Ni and Fe elements is observed in XPS. The reason for this is that the oxidation of Ni and Fe elements on the surface of wood composites with a depth of several nanometers is unavoidable at atmospheric conditions [38].

### 3.5. Magnetism Analysis of Composites

Figure 7a shows the magnetic hysteresis loops of the modified wood composite at room temperature. It is obvious that all the composites exhibit a typical hysteretic behavior. The saturation magnetization varies with the applied magnetic field. Magnetic wood composites with FeNi_3_ loading of 12, 15 and 18 wt % possess saturation magnetizations of 6.3, 8.1 and 10.8 emu/g, respectively. This result points out that higher FeNi_3_ loading in the wood matrix would better increase the saturation of magnetization. Furthermore, the saturation of magnetization of FeNi_3_ modified wood composites is higher than iron oxide-modified thermoplastic wood materials [39] and CoFe_2_O_4_-modified wood composites [40]. Further, it is observed that the coercive forces of modified wood are about 45 Oe, which indicates that these modified wood composites display a soft magnetism behavior [41]. However, the saturation of magnetization of the FeNi_3_ modified wood composites is obviously lower than the FeNi_3_ nanoparticles (110 emu/g and 48.48 emu/g, respectively) [23,24]. The reason for this is that the wood leads to a magnetically dead layer [42].

Since the native biomaterial wood possesses intrinsic anisotropy [43], whether the MW–18 wt % wood fiber directions were parallel or vertical to the applied magnetics direction was compared. As shown in Figure 7b, when the wood fiber was parallel to the external magnetic field, the magnetic hysteresis loops were significantly steeper, suggesting that it is easier to be magnetized this way. Because the magnetic particles were mainly deposited along the longitudinal fiber growth direction, the fiber orientation will influence the magnetic properties [44]. Thus, the result indicates that the magnetism of FeNi_3_ nanoparticle modified wood shows directional dependence, which is similar to the ferrite loaded wood reported before [45].

Figure 8 shows the excellent magnetic property of the modified wood. The magnetic wood can be firmly attracted and easily lifted by a permanent magnet, as seen in Figure 8a. The wood specimen can be observed floating on the surface of the water in Figure 8b. The wood can overcome the buoyancy of the water and be attracted to the bottom of the cup by using the external magnet, as demonstrated in Figure 8c.

## 4. Conclusions

In summary, magnetic wood composites with different saturations of magnetization were fabricated by in situ chemosynthesis. The FeNi_3_ nanoparticles were successfully synthesized in the wood matrix. Magnetic wood composites exhibit better a saturation of magnetization, with an increase in FeNi_3_ content. The fiber orientation in the wood structure has a certain influence on the magnetic characteristics. The presented approach provides a method for producing inexpensive, lightweight and soft magnetic wood composites.

## Figures and Tables

**Figure 1 polymers-11-00421-f001:**
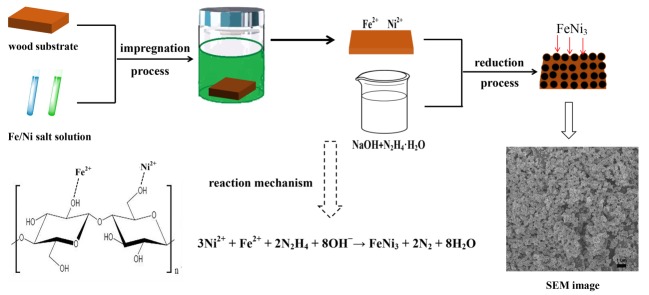
Schematic of preparation for FeNi_3_ modified wood composites.

**Figure 2 polymers-11-00421-f002:**
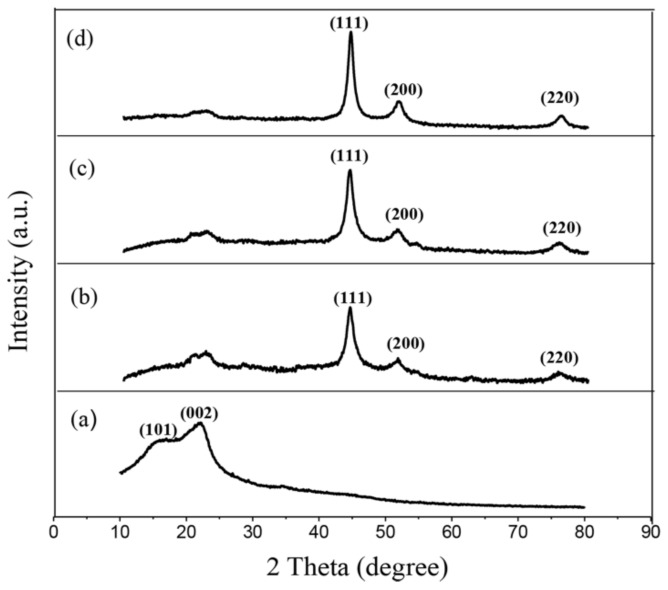
X-ray diffraction (XRD) patterns of unmodified and modified wood. (**a**) Unmodified wood, (**b**) MW–12 wt %, (**c**) MW–15 wt % and (**d**) MW–18 wt %.

**Figure 3 polymers-11-00421-f003:**
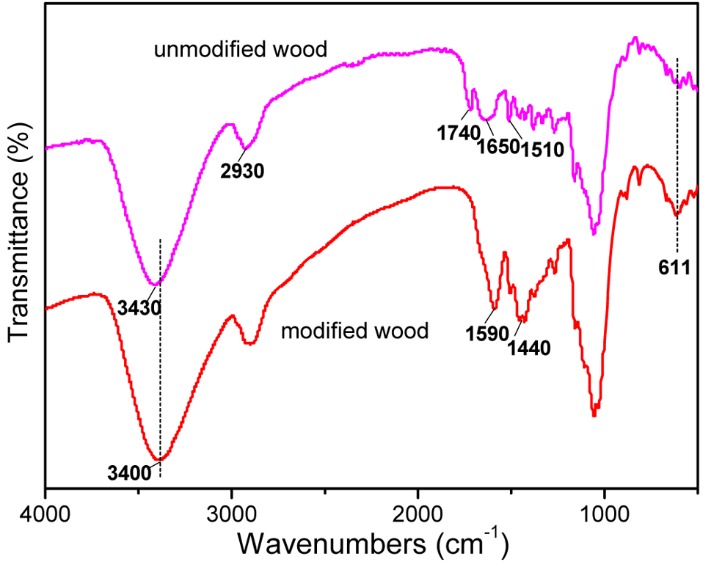
Fourier transform infrared (FTIR) spectroscopy of unmodified wood and MW–18 wt %.

**Figure 4 polymers-11-00421-f004:**
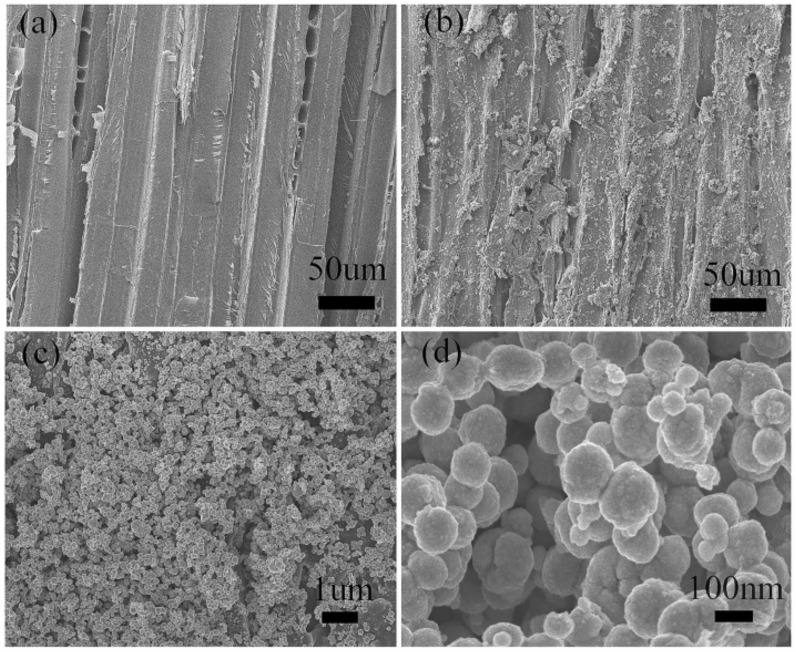
Scanning electron microscopy (SEM) images of unmodified wood and MW–18 wt %. (**a**) Unmodified wood (2000×), (**b**) modified wood (2000×), (**c**) FeNi_3_ nanoparticles (100,000×), (**d**) FeNi_3_ nanoparticles (1,000,000×).

**Figure 5 polymers-11-00421-f005:**
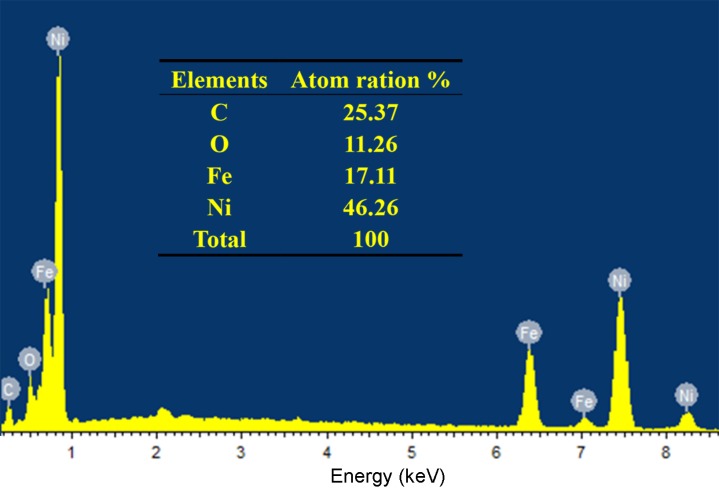
Energy-dispersive X-ray (EDX) spectrum of MW–18 wt %.

**Figure 6 polymers-11-00421-f006:**
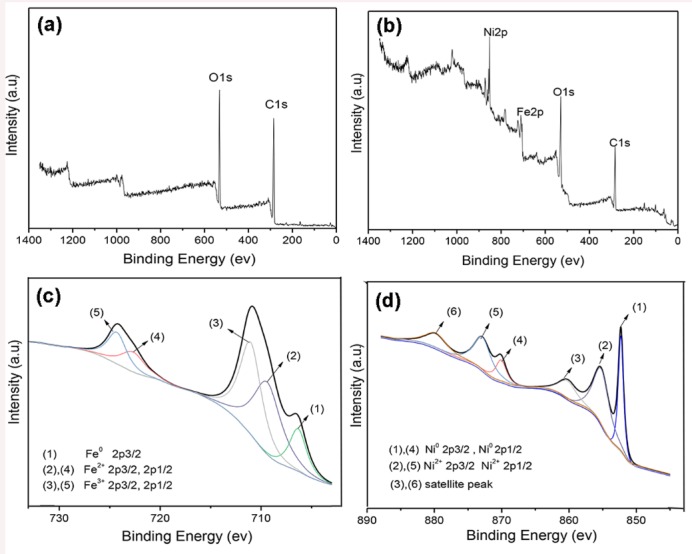
X-ray photoelectron spectroscopy (XPS) spectra of unmodified wood and MW–18 wt %. (**a**) Unmodified wood, (**b**) MW–18 wt %, (**c**) XPS spectrum of Fe 2p and (**d**) XPS spectrum of Ni 2p.

**Figure 7 polymers-11-00421-f007:**
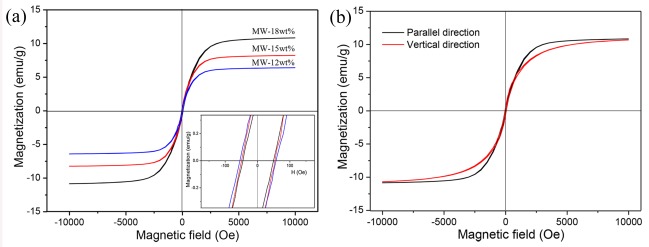
Magnetization curves of modified wood composites. (**a**) MW–12 wt %, MW–15 wt % and MW–18 wt %. (**b**) MW–18 wt % at parallel and vertical direction.

**Figure 8 polymers-11-00421-f008:**
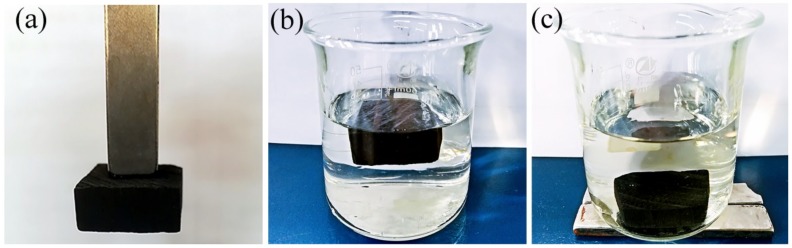
Digital images of the modified wood with an excellent magnetic property. (**a**) Magnetic wood can be firmly attracted by permanent magnet. (**b**) The specimen floating on the surface of the water. (**c**) The specimen can be attracted to the bottom of the cup by an external magnet.

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
