# Peer review of "Magnetic Properties of FeNi3 Nanoparticle Modified Pinus radiata Wood Nanocomposites"

_polymers, 2019, doi:10.3390/polym11030421_

Round 1
Reviewer 1 Report
General assessment:
The paper concern preparation of magnetic nanocomposite made of wood and FeNi3. The topic is interesting due to importance of nanocomposite materials. The work is properly designed but results are rather routine and novelty is rather modest. Application of this compound for wood modification is the most interesting in this work. Magnetic properties of this nanocomposite have been studied but they are presented without any reference to literature data for FeNi3. Please, complete this part with additional data. Moreover, I doubt if according current version of this paper anyone would be able to repeat results due to missing experimental data. Therefore, I would appreciate significantly enlarged experimental part. The careful English review is also required.
Introduction is well written but should provide more information. Please, enlarge it.
Experimental:
Lines 50-51: the description is very short and incomplete (e.g. time, temperature, other factors important for the process) making this experiment extremely difficult to repeat.
Line 59-60: “of 0.6 M, 0.8 M and 1.0 M for mixed solution A, B and C, respectively.” what do these concentrations mean? Are they related to iron or nickel compounds or to final concentration of the product or to something else? Hence, please indicate clearly what A, B and C are.
Language:
Line 28: “can as a natural porosity scaffold to combine with…” should be “can combine a natural porosity scaffold with...”
Line 30: “by filling” I suppose it would be better „by dispersing”
Line 35: “woody” should be “wood”
Line 61: “Firstly, the immersed process of wood under vacuum (0.09 MPa) for 2 h.” no verb. Please convert it into sentence. 0.09 MPa is not at all a vacuum, it is only slightly lowered pressure!
Line 71 and 77: “schematic” should be “the scheme”
Line 100: “destroy” it is better “destroying”
Line 120: “were occurred” should be removed
Line 122: “are occurred” should be “occurred”
Line 136: “by situ” should be “by in situ”
Author Response
Response to Reviewers
Firstly I would like to take this opportunity to thank your carefully reading and your constructive comments and suggestions.
Answer to Reviewer #1:
Q1-1 The paper concern preparation of magnetic nanocomposite made of wood and FeNi3. The topic is interesting due to importance of nanocomposite materials. The work is properly designed but results are rather routine and novelty is rather modest. Application of this compound for wood modification is the most interesting in this work. Magnetic properties of this nanocomposite have been studied but they are presented without any reference to literature data for FeNi3. Please, complete this part with additional data. Moreover, I doubt if according current version of this paper anyone would be able to repeat results due to missing experimental data. Therefore, I would appreciate significantly enlarged experimental part. The careful English review is also required.
A1-1. Answer to reviewer:
We appreciate your suggestion very much. In order to add the literature data for FeNi3, the introduction was rewritten. The previous contents “When the molar ratio of Fe to Ni is 1/3, FeNi3 intermetallic compound can be obtained [23]. The FeNi3 nanoparticles modified wood composites have not been reported until now. In this study, FeNi3 particles were formed in the cell cavity of wood by a simple in situ synthesis.” was changed into “When the molar ratio of Fe to Ni is 1/3, FeNi3 intermetallic compound can be obtained. Lu et al. reported that ferromagnetic FeNi3 nanoparticles have been successfully synthesized via hydrazine hydrate reduction in aqueous solution at room temperature. Furthermore, the saturation magnetization reaches about 110 emu/g [23]. In addition, Yuan et al. reported that FeNi3 alloy were prepared using hydrazine hydrate as a reducing agent in strong alkaline media. The saturation magnetization of FeNi3 alloy is 48.48 emu/g [24]. However, FeNi3 nanoparticles modified wood composites have not been reported until now. Hydrazine hydrate is a moderate reducing agent with low-cost and has been proved to be good alternative to produce FeNi3 metallic particles. Therefore, in this study, FeNi3 nanoparticles were formed in the cell cavity of wood using hydrazine hydrate as a reducing agent by a simple in situ synthesis.” in the revised manuscript. (see lines 43-57)
The relevant literatures are as follows:
23. Lu, X.; Liang, G.; Zhang, Y. Synthesis and characterization of magnetic FeNi3 particles obtained by hydrazine reduction in aqueous solution. Mater. Sci. Eng., B 2007, 139, 124-127, (doi:https://doi.org/10.1016/j.mseb.2007.01.055). (see lines 333-335)
24. Yuan, M.L.; Tao, J.H.; Yu, L.; Song, C.; Qiu, G.Z.; Li, Y.; Xu, Z.H. Synthesis and magnetic properties of Fe–Ni alloy nanoparticles obtained by hydrothermal reaction. Advanced Materials Research 2011, 239-242, 748-753, (doi:10.4028/www.scientific.net/AMR.239-242.748). (see lines 336-338)
Furthermore, in the “3.5. Magnetism Analysis of Composites” section, the new added contents are as follows: “However, the saturation magnetization of FeNi3 modified wood composites are obviously lower than FeNi3 nanoparticles (110 emu/g and 48.48 emu/g) [23, 24]. The reason is that wood leads to a magnetically dead layer [42].” (see lines 233-235)
The new added literature is as follows:
42. Wu, H.; Zhang, R.; Liu, X.X.; Lin, D.D.; Pan, W. Electrospinning of Fe, Co, and Ni nanofibers: Synthesis, assembly, and magnetic properties. Chemistry Of Materials 2007, 19, 3506-3511. (see lines 386-387)
We appreciate your suggestion very much. The experimental part is described in detail. The previous contents “Wood (Pinus radiata) was treated as a block of 20 mm × 20 mm × 10 mm and dried in a vacuum oven (DHG-9203A, Shanghai , China) to constant weight.” was changed into “Radiata pine specimens were cut into blocks of 20 mm (longitudinal) × 20 mm (radial) × 10 mm (tangential) and dried in a vacuum oven (DHG-9203A, Shanghai , China) at 60 °C for 12 h to constant weight.” in the revised manuscript. (see lines 61-63)
The previous words “hydrate hydrazine (N2H4·H2O)” was changed into “hydrate hydrazine (N2H4·H2O, 80%)” in the revised manuscript. (see line 65)
The previous contents “The mixture of (NH4)2Fe(SO4)2·6H2O and NiSO4·6H2O with a molar ratio of 1/3 was dissolved in distilled water to form a homogeneous solution with the concentration of 0.6 M, 0.8 M and 1.0 M for mixed solution A, B and C, respectively. Wood specimens were entirely dipped into A, B, and C solution, respectively. Firstly, the immersed process of wood under vacuum (0.09 MPa) for 2 h. Subsequently, the above pretreated wood samples were rapidly soaked into alkaline hydrate hydrazine solution with a mole concentration of 3 M at 70 °C for 6 h, the formed nanoparticles was labeled as FeNi3.” was changed into “The mixture of (NH4)2Fe(SO4)2·6H2O and NiSO4·6H2O with a molar ratio of 1/3 was dissolved in 50 mL distilled water to form three homogeneous solution A, B and C, respectively. Mixed solution A with the concentration of 0.6 mol/L contains 2.94 g (NH4)2Fe(SO4)2·6H2O (0.0075 mol) and 5.91 g NiSO4·6H2O (0.0225 mol); Mixed solution B with the concentration of 0.8 mol/L contains 3.92 g (NH4)2Fe(SO4)2·6H2O (0.01 mol) and 7.88 g NiSO4·6H2O (0.03 mol); Mixed solution C with the concentration of 1.0 mol/L contains 4.90 g (NH4)2Fe(SO4)2·6H2O (0.0125 mol) and 9.86 g NiSO4·6H2O (0.0375 mol); Firstly, wood specimens were entirely dipped into A, B, and C solution under lowered pressure of 0.09 MPa for 2 h, respectively. NaOH (2 g) and hydrazine hydrate (9.0 mL) were added in 50 mL distilled water to form alkaline hydrate hydrazine solution. Subsequently, the above pretreated wood samples were rapidly soaked into alkaline hydrate hydrazine solution with a mole concentration of 3 mol/L at 70 °C for 6 h, the formed nanoparticles was labeled as FeNi3.” in the revised manuscript. (see lines 70-80)
Q1-2 Introduction is well written but should provide more information. Please, enlarge it.
A1-2. Answer to reviewer:
We appreciate your suggestion very much. In order to add the literature data for FeNi3, the introduction was rewritten. The previous contents “When the molar ratio of Fe to Ni is 1/3, FeNi3 intermetallic compound can be obtained [23]. The FeNi3 nanoparticles modified wood composites have not been reported until now. In this study, FeNi3 particles were formed in the cell cavity of wood by a simple in situ synthesis.” was changed into “When the molar ratio of Fe to Ni is 1/3, FeNi3 intermetallic compound can be obtained. Lu et al. reported that ferromagnetic FeNi3 nanoparticles have been successfully synthesized via hydrazine hydrate reduction in aqueous solution at room temperature. Furthermore, the saturation magnetization reaches about 110 emu/g [23]. In addition, Yuan et al. reported that FeNi3 alloy were prepared using hydrazine hydrate as a reducing agent in strong alkaline media. The saturation magnetization of FeNi3 alloy is 48.48 emu/g [24]. However, FeNi3 nanoparticles modified wood composites have not been reported until now. Hydrazine hydrate is a moderate reducing agent with low-cost and has been proved to be good alternative to produce FeNi3 metallic particles. Therefore, in this study, FeNi3 nanoparticles were formed in the cell cavity of wood using hydrazine hydrate as a reducing agent by a simple in situ synthesis.” in the revised manuscript. (see lines 43-57)
The relevant literatures are as follows:
23. Lu, X.; Liang, G.; Zhang, Y. Synthesis and characterization of magnetic FeNi3 particles obtained by hydrazine reduction in aqueous solution. Mater. Sci. Eng., B 2007, 139, 124-127, (doi:https://doi.org/10.1016/j.mseb.2007.01.055). (see lines 333-335)
24. Yuan, M.L.; Tao, J.H.; Yu, L.; Song, C.; Qiu, G.Z.; Li, Y.; Xu, Z.H. Synthesis and magnetic properties of Fe–Ni alloy nanoparticles obtained by hydrothermal reaction. Advanced Materials Research 2011, 239-242, 748-753, (doi:10.4028/www.scientific.net/AMR.239-242.748). (see lines 336-338)
Q1-3 Lines 50-51: the description is very short and incomplete (e.g. time, temperature, other factors important for the process) making this experiment extremely difficult to repeat.
A1-3. Answer to reviewer:
Thank you for your suggestion. The previous sentence “Wood (Pinus radiata) was treated as a block of 20 mm × 20 mm × 10 mm and dried in a vacuum oven (DHG-9203A, Shanghai , China) to constant weight.” was changed into “Radiata pine specimens were cut into blocks of 20 mm (longitudinal) × 20 mm (radial) × 10 mm (tangential) and dried in a vacuum oven (DHG-9203A, Shanghai , China) at 60 °C for 12 h to constant weight.” in the revised manuscript. (see lines 61-63)
Q1-4 Line 59-60: “of 0.6 M, 0.8 M and 1.0 M for mixed solution A, B and C, respectively.” what do these concentrations mean? Are they related to iron or nickel compounds or to final concentration of the product or to something else? Hence, please indicate clearly what A, B and C are.
A1-4. Answer to reviewer:
We appreciate your suggestion very much. The previous content “The mixture of (NH4)2Fe(SO4)2·6H2O and NiSO4·6H2O with a molar ratio of 1/3 was dissolved in distilled water to form a homogeneous solution with the concentration of 0.6 M, 0.8 M and 1.0 M for mixed solution A, B and C, respectively.” has been changed into “The mixture of (NH4)2Fe(SO4)2·6H2O and NiSO4·6H2O with a molar ratio of 1/3 was dissolved in 50 mL distilled water to form three homogeneous solution A, B and C, respectively. Mixed solution A with the concentration of 0.6 mol/L contains 2.94 g (NH4)2Fe(SO4)2·6H2O (0.0075 mol) and 5.91 g NiSO4·6H2O (0.0225 mol); Mixed solution B with the concentration of 0.8 mol/L contains 3.92 g (NH4)2Fe(SO4)2·6H2O (0.01 mol) and 7.88 g NiSO4·6H2O (0.03 mol); Mixed solution C with the concentration of 1.0 mol/L contains 4.90 g (NH4)2Fe(SO4)2·6H2O (0.0125 mol) and 9.86 g NiSO4·6H2O (0.0375 mol);” in the revised manuscript. (see lines 70-76)
Q1-5 Line 28: “can as a natural porosity scaffold to combine with…” should be “can combine a natural porosity scaffold with...”
A1-5 . Answer to reviewer:
Thank you for your suggestion. The previous sentence “Furthermore, wood with micro porous structure can as a natural porosity scaffold to combine with inorganic/organic compounds.” was changed into “Furthermore, wood with micro porous structure can combine a natural porosity scaffold with inorganic/organic compounds” in the revised manuscript. (see lines 28-29)
Q1-6 Line 30: “by filling” I suppose it would be better „by dispersing”
A1-6 . Answer to reviewer:
Thank you for your suggestion. The previous sentence “Therefore, many efforts have been devoted to creating functionalized wood by filling inorganic and organic components into the wood matrix in order to endow these wood–based materials with new properties,” was changed into “Therefore, many efforts have been devoted to creating functionalized wood by dispersing inorganic and organic components into the wood matrix in order to endow these wood–based materials with new properties,” in the revised manuscript. (see lines 29-31)
Q1-7 Line 35: “woody” should be “wood”
A1-7 . Answer to reviewer:
Thank you for your suggestion. The previous word “woody” was changed into “wood” in the revised manuscript. (see line 35)
Q1-8 Line 61: “Firstly, the immersed process of wood under vacuum (0.09 MPa) for 2 h.” no verb. Please convert it into sentence. 0.09 MPa is not at all a vacuum, it is only slightly lowered pressure!
A1-8. Answer to reviewer:
We appreciate your suggestion very much. The previous content “Wood specimens were entirely dipped into A, B, and C solution, respectively. Firstly, the immersed process of wood under vacuum (0.09 MPa) for 2 h.” has been changed into “Firstly, wood specimens were entirely dipped into A, B, and C solution under lowered pressure of 0.09 MPa for 2 h, respectively.” in the revised manuscript. (see lines 76-77)
Q1-9 Line 71 and 77: “schematic” should be “the scheme”
Line 100: “destroy” it is better “destroying”
A1-9. Answer to reviewer:
Thank you for your correction. The previous sentence “Furthermore, experimental schematic for the preparation of FeNi3 nanoparticles modified wood is displayed in Figure 1.” was changed into “Furthermore, the experimental scheme for the preparation of FeNi3 nanoparticles modified wood is displayed in Figure 1.” in the revised manuscript. (see lines 89-91)
The previous word “destroy” has been changed into “destroying” in the revised manuscript. The revised sentence is as follows: “For all of the modified wood, these characteristic peaks of pristine wood are distinctly weakened in Fig. 2(b)–(d), which is due to the deposited metal particles destroying the crystalline of the cellulose, resulting in a decrease in the crystalline of the cellulose.” (see lines 148-151)
Q1-10 Line 120: “were occurred” should be removed
A1-10. Answer to reviewer:
Thank you for your correction. The previous words “were occurred” was removed in the revised manuscript. The revised sentence is as follows: “The bands at around 1730 cm–1 has disappeared because of the degradation of acetyl groups in hemicelluloses by magnetic treatment [30].” (see lines 170-171)
Q1-11 Line 122: “are occurred” should be “occurred”
A1-11. Answer to reviewer:
Thank you for your correction. The previous words “are occurred” has been changed into “occurred” in the revised manuscript. The revised sentence is as follows: “Meanwhile, the O–H stretching absorption band at 3430 cm−1 shift to the low wavenumber with 3400 cm–1, confirming that the interactions occurred between the OH groups of wood matrix and the formed FeNi3 nanoparticles through hydrogen bonds [31].” (see lines 172-174)
Q1-12 Line 136: “by situ” should be “by in situ”
A1-12. Answer to reviewer:
Thank you for your correction. The previous words “by situ” has been changed into “by in situ” in the revised manuscript. The revised sentence is as follows: “The above SEM tests confirm that FeNi3 nanoparticles are imbedded into the inner cavity of wood by in situ chemosynthesis.” (see lines 192-194)

Reviewer 2 Report
This manuscript describes the preparation of a intermetallic compound prepared by in situ soaking of the natural wood into a metallic ion solution. The preparation method is well described, however I wonder about the reproducibility of the loaded NPs wood composite. Are these NPs distributed only on the surface of the wood material or are they placed inside the wood porosity too? Authors could analyze the edge of a wood cut to discover if the NPs are also placed inside the wood and if they are homogeneous distributed, since it affects to the final properties of the composite.
Have the authors corroborate the reproducibility of the NPs loading? They do not mention any repetition experiment. Usually, in situ preparation processes present poor reproducibility.
Have the authors done any strength, compression and/or tension wood test to see the effect of the temperature and high alkaline treatment in the treated wood? Authors do not mention any clear application, however these properties are important in a material as wood to be taken into consideration.
In the FTIR spectrum of the nanocomposite, it is not clear the region between 400 and 700 cm-1, where it could be observed any signal referred to the O-Fe or O-Ni ionic bonds.
EDAX results show a very high proportion of Ni and Fe versus C and O, that are the wood components. That means that EDAX analysis have been done basically from the NPs layer, but it would be interesting to compare these results with the EDAX analysis of the unmodified wood to have an idea of the proportion of NPs in the wood and see if the NPs percentage obtained by difference weight is the same.
Author Response
Response to Reviewers
Firstly I would like to take this opportunity to thank your carefully reading and your constructive comments and suggestions.
Answer to Reviewer #2:
Q2-1 This manuscript describes the preparation of a intermetallic compound prepared by in situ soaking of the natural wood into a metallic ion solution. The preparation method is well described, however I wonder about the reproducibility of the loaded NPs wood composite. Are these NPs distributed only on the surface of the wood material or are they placed inside the wood porosity too? Authors could analyze the edge of a wood cut to discover if the NPs are also placed inside the wood and if they are homogeneous distributed, since it affects to the final properties of the composite.
A2-1. Answer to reviewer:
We appreciate your comment very much. The mixture of (NH4)2Fe(SO4)2·6H2O and NiSO4·6H2O is homogeneous aqueous solution. Firstly, wood specimens were entirely dipped into aqueous solution of (NH4)2Fe(SO4)2·6H2O and NiSO4·6H2O under lowered pressure of 0.09 MPa. Because the mixture of (NH4)2Fe(SO4)2·6H2O and NiSO4·6H2O is an aqueous solution, Fe2+ and Ni2+ can easily enter into the internal cavity of wood. Subsequently, the above pretreated wood samples were rapidly soaked into alkaline hydrate hydrazine solution with a mole concentration of 3 mol/L at 70 °C for 6 h.
The alkaline hydrate hydrazine solution is homogeneous aqueous solution of sodium hydroxide (NaOH) and hydrate hydrazine (N2H4·H2O). Therefore, hydrazine hydrate as a reducing agent can be easily immersed in the internal cavity of wood. Furthermore, the reaction temperature is very mild (70 °C). Reaction conditions are easy to control. We repeated this experiment, and modified wood composite MW–18 wt% has been reproduced. The magnetic hysteresis loop of new modified wood composite MW–18 wt% is shown in Figure A1.
Figure A1. Magnetic hysteresis loop of new modified wood composite MW–18 wt%.
The saturation magnetization of new modified wood composite MW–18 wt% is 11.0 emu/g, which is very close to 10.8 emu/g of previous MW–18 wt% in the manuscript. Therefore, the reproducibility of wood composites is successful. Magnetic wood composites with FeNi3 loading of 12, 15, and 18 wt% possess the saturation magnetization of 6.3, 8.1 and 10.8 emu/g, respectively. The magnetic hysteresis loops of previous modified wood composite in the manuscript are as follows:
Figure 7 (a) . Magnetization curves of modified wood composites. (a) MW–12wt%, MW–15wt% and MW–18 wt%.
Radiata pine specimens were cut into blocks of 20 mm (longitudinal) × 20 mm (radial) × 10 mm (tangential). The FeNi3 nanoparticles distributed not only on the surface of the wood material,but also placed inside the wood porosity too. Figure 4(a) and (b) show the SEM images of longitudinal cross-sections of pristine wood and MW–18 wt% wood composite, respectively. The longitudinal cross-sections show the morphology inside the wood porosity. From Fig. 4(a), it can be seen that the unmodified wood have micro–grooved structures and smooth lumen walls inside the wood porosity. However, modified wood composite MW–18 wt% show that micro–grooved structures of original wood have been covered by modified particles in Fig. 4(b). It reveals that FeNi3 particles are adhering to the lumen walls inside the wood porosity.
Figure 4. SEM images of unmodified wood and MW–18 wt%. (a) unmodified wood (2000×),
(b) modified wood (2000×).
Q2-2 Have the authors corroborate the reproducibility of the NPs loading? They do not mention any repetition experiment. Usually, in situ preparation processes present poor reproducibility.
A2-2. Answer to reviewer:
We appreciate your comment very much. The modified wood composite MW–18 wt% has been reproduced. The saturation magnetization of new modified wood composite MW–18 wt% is 11.0 emu/g, which is very close to 10.8 emu/g of previous MW–18 wt% in the manuscript. Therefore, the reproducibility of wood composites is successful. The magnetic hysteresis loop of new modified wood composite MW–18 wt% at room temperature is as follows:
Figure A1. Magnetic hysteresis loop of new modified wood composite MW–18 wt%.
Q2-3 Have the authors done any strength, compression and/or tension wood test to see the effect of the temperature and high alkaline treatment in the treated wood? Authors do not mention any clear application, however these properties are important in a material as wood to be taken into consideration.
A2-3. Answer to reviewer:
Thank you for your suggestion. Wood specimens were heated at 130 °C and 270 °C for 3 h in a muffle furnace, respectively. Maximal degradation-rate temperatures (Tmax) of heated wood at 130 °C is 360 °C, which is same with Tmax (360 °C) of pristine wood. However, Tmax of heated wood at 270 °C is 340 °C, which is distinctly lower than that of Tmax of pristine wood.
DTG curve of treated wood at room temperature is shown in Figure A2.
Figure A2. DTG curve of treated wood at room temperature.
DTG curve of treated wood at 270 °C is shown in Figure A3.
Figure A3. TG and DTG curves of heated wood at 270 °C.
In addition, the wood samples were impregnated in a 2.5 mol·L-1 NaOH solution with rapid stirring at 100 °C for 8 h in a beaker. The result indicated that partial lignin was removed from wood samples under high alkali treatment. Thus, the mechanical properties of wood will be reduced. The thermal stability and composition of wood are changed under high temperature and high alkali treatment. Therefore, strength, compression and/or tension of wood will be effected by high temperature and high alkali treatment.
Q2-4 In the FTIR spectrum of the nanocomposite, it is not clear the region between 400 and 700 cm-1, where it could be observed any signal referred to the O-Fe or O-Ni ionic bonds.
A2-4. Answer to reviewer:
We appreciate your comment very much. A new sentence “A new characteristic band at 611 cm-1 is attributed to O-Fe and O-Ni stretching modes [32, 33].” was added to the revised version. (see lines 174-175)
The added references are as follows:
[32] Bheki Magagula, N.N., Walter W. Focke. Mn2al-ldh- and co2al-ldh-stearate as photodegradants for ldpe film. 2009, 94, 947-954, (doi:10.1016/j.polymdegradstab.2009.03.007). (see lines 358-359)
[33] Alekseeva, T.; Prevot, V.; Sancelme, M.; Forano, C.; Besse-Hoggan, P. Enhancing atrazine biodegradation by pseudomonas sp. Strain adp adsorption to layered double hydroxide bionanocomposites. J. Hazard. Mater. 2011, 191, 126-135, (doi:10.1016/j.jhazmat.2011.04.050). (see lines 360-362 )
The FTIR spectra of unmodified wood and MW–18 wt% in the revised manuscript are as follows: (see lines 178-179 )
Figure 3. FTIR spectra of unmodified wood and MW–18 wt%.
Q2-5 EDAX results show a very high proportion of Ni and Fe versus C and O, that are the wood components. That means that EDAX analysis have been done basically from the NPs layer, but it would be interesting to compare these results with the EDAX analysis of the unmodified wood to have an idea of the proportion of NPs in the wood and see if the NPs percentage obtained by difference weight is the same.
A2-5. Answer to reviewer:
Thank you for your suggestion. Indeed, this EDAX analysis have been done basically from the FeNi3 particles layer. The selected area for EDAX analysis is as follows:
EDAX image and the atomic ratio of elements are as follows:
Both EDAX analysis and wide scan XPS spectra can characterize the composition of elements. Fig. 6(a) show wide scan XPS spectra for unmodified wood in the manuscript. C and O are major elements of unmodified wood.
Figure 6 (a) . XPS spectra of unmodified wood.
From the following peak table, it can be seen that At.% of C (46.69 %) and O (53.31 %) elements for the unmodified wood. However, neither wide scan XPS spectra nor EDAX analysis can measure the hydrogen content. Hydrogen is also the main element of wood. Therefore, although the main elements of unmodified wood, carbon and oxygen, have been tested, it is still impossible to calculate the percentage of FeNi3 alloy in wood quality.

Reviewer 3 Report
The authors report the magnetic properties of FeNi3 nanoparticles embedded in Pinus radiata wood. The samples were investigated by various techniques. The work is interesting and the synthesized system has potential applications. Paper is well written. I recommend this paper to be accepted in Polymers.
Author Response
I would like to take this opportunity to thank your carefully reading and your encourage. Thank you very much!
Round 2
Reviewer 1 Report
I appreciate significant effort made by Authors to improve their paper. However, I do not feel that this paper is extremely interesting to readers of Polymers also due to the limited level of novelty. Nevertheless, due to the mentioned effort and significantly bigger amount of information and details allowing for repetition of these results I suggest to accept this paper.
In the manuscript on p. 7 Fig. 6 is repeated twice. please, remove one copy.
Reviewer 2 Report
Thank you for the response to all my comments, I can understand much better all the scientific results and the new version of the manuscript is good for publication.